# The Kv2.2 channel mediates the inhibition of prostaglandin E2 on glucose-stimulated insulin secretion in pancreatic β-cells

Chengfang Pan[1,2†], Ying Liu[3†], Liangya Wang[1,2], Wen-Yong Fan[1], Yunzhi Ni[3], Xuefeng Zhang[1], Di Wu[3], Chenyang Li[1], Jin Li[1], Zhaoyang Li[4*], Rui Liu[3*], Changlong Hu[1,2*]

[1]School of Life Sciences, Fudan University, Shanghai, China; [2]International Human Phenome Institute (Shanghai), Shanghai, China; [3]Department of Endocrinology and Metabolism, Huashan Hospital, Fudan University, Shanghai, China; [4]State Key Laboratory of Medical Neurobiology and MOE Frontiers Center for Brain Science, and the Institutes of Brain Science, Fudan University, Shanghai, China

*For correspondence:
lzy@fudan.edu.cn (ZL);
fd_ruiliu@fudan.edu.cn (RL);
clhu@fudan.edu.cn (CH)

†These authors contributed equally to this work

## eLife Assessment

The study presents **valuable** findings on the molecular mechanisms of glucose-stimulated insulin secretion from pancreatic islets, focusing on the main regulatory elements of the signaling pathway in physiological conditions. While the evidence supporting the conclusions is **solid**, the study can be strengthened by the use of a beta cell line or knockout mice. The work will be of interest to cell biologists and biochemists working on diabetes.

**Abstract** Prostaglandin E2 (PGE2) is an endogenous inhibitor of glucose-stimulated insulin secretion (GSIS) and plays an important role in pancreatic β-cell dysfunction in type 2 diabetes mellitus (T2DM). This study aimed to explore the underlying mechanism by which PGE2 inhibits GSIS. Our results showed that PGE2 inhibited Kv2.2 channels via increasing PKA activity in HEK293T cells overexpressed with Kv2.2 channels. Point mutation analysis demonstrated that S448 residue was responsible for the PKA-dependent modulation of Kv2.2. Furthermore, the inhibitory effect of PGE2 on Kv2.2 was blocked by EP2/4 receptor antagonists, while mimicked by EP2/4 receptor agonists. The immune fluorescence results showed that EP1–4 receptors are expressed in both mouse and human β-cells. In INS-1(832/13) β-cells, PGE2 inhibited voltage-gated potassium currents and electrical activity through EP2/4 receptors and Kv2.2 channels. Knockdown of *Kcnb2* reduced the action potential firing frequency and alleviated the inhibition of PGE2 on GSIS in INS-1(832/13) β-cells. PGE2 impaired glucose tolerance in wild-type mice but did not alter glucose tolerance in *Kcnb2* knockout mice. Knockout of *Kcnb2* reduced electrical activity, GSIS and abrogated the inhibition of PGE2 on GSIS in mouse islets. In conclusion, we have demonstrated that PGE2 inhibits GSIS in pancreatic β-cells through the EP2/4-Kv2.2 signaling pathway. The findings highlight the significant role of Kv2.2 channels in the regulation of β-cell repetitive firing and insulin secretion, and contribute to the understanding of the molecular basis of β-cell dysfunction in diabetes.

## Introduction

Prostaglandin E2 (PGE2) is the major prostaglandin formed in pancreatic islets and is closely related to islet β-cell dysfunction (*Meng et al., 2006*; *Oshima et al., 2006*; *Vennemann et al., 2012*). As an endogenous inhibitor of glucose-stimulated insulin secretion (GSIS), PGE2 plays an important role in type 2 diabetes mellitus (T2DM) (*Carboneau et al., 2017*). PGE2 has been demonstrated to inhibit GSIS in both mouse models of T2DM and in pancreatic islets obtained from human organ donors with T2DM (*Kimple et al., 2013*; *Neuman et al., 2017*; *Parazzoli et al., 2012*). Recent studies have shown that plasma levels of PGE2 are correlated with T2DM status, and it has the potential to be a marker of T2DM status (*Fenske et al., 2022*; *Truchan et al., 2021*). However, the underlying mechanism for the PGE2 inhibition of GSIS is not fully understood.

PGE2 functions through the activation of four specific G protein-coupled receptor subtypes, termed EP1–4. The EP1 receptor couples to $G_q$ and causes an intracellular $Ca^{2+}$ increase. The EP2 and EP4 receptors couple to $G_s$ to increase intracellular cAMP formation, while the EP3 receptor couples to $G_i$ to decrease intracellular cAMP production (*Sugimoto and Narumiya, 2007*). The mRNA expression of all four EP receptors is found in rat, mouse, and human pancreatic islets (*Bramswig et al., 2013*; *Tran et al., 2002*; *Vennemann et al., 2012*). Among all EP receptors, the EP3 receptor has attracted widespread attention due to its high expression on β-cells in diabetic mouse models (elevated more than 40 times compared to nondiabetic mice) (*Kimple et al., 2013*). Currently, it is considered the primary effector for PGE2 in GSIS. Islets from diabetic BTBR mice exhibit increased GSIS when treated with the EP3 antagonist L-798106 and show decreased GSIS after treated with the EP3 agonist PGE1 (*Shridas et al., 2014*). However, increasing evidence suggests that EP3 plays a role only in insulin secretion under T2DM (*Carboneau et al., 2017*). In human islets from nondiabetic donors, the EP3 antagonist L-798106 does not affect GSIS but improves insulin secretion in islets from donors with T2DM (*Kimple et al., 2013*). *Ceddia et al., 2016* reported that the EP3 antagonist DG-041 does not affect GSIS in islets from nondiabetic human or wild-type mice on a chow diet, and that global gene knockout of EP3 in mice does not alter GSIS. The impact of the PGE2 signaling pathway on nondiabetic β-cell function remains unclear. Moreover, the roles of EP1, EP2, and EP4 receptors in GSIS are rarely reported.

Pancreatic β-cells are electrical excitable. GSIS is associated with a complex electrical activity, which is regulated by various voltage-gated plasmalemmal ion channels (*Braun et al., 2008*; *Yang et al., 2014*). The Kv2 channel family, which consists of Kv2.1 and Kv2.2, plays a crucial role in modulating neural excitability (*Liu and Bean, 2014*). The effect of Kv2.1 channels on insulin secretion has been well studied. In rodents, Kv2.1 is recognized as the primary facilitator of delayed rectifier potassium currents in β-cells. Suppressing Kv2.1 results in heightened glucose-stimulated membrane potential amplitude, consequently enhancing GSIS in mouse pancreatic β-cells (*Jacobson et al., 2007*; *MacDonald et al., 2002*). However, Kv2.1 inhibitor stromatoxin shows little effect on human β-cell electrical function or insulin secretion (*Braun et al., 2008*), despite the presence of Kv2.1 protein in human islets (*Tamarina et al., 2005*). Increasing evidence suggests that Kv2.1 regulates insulin secretion through channel clusters independently of its electrical function (*Dai et al., 2012*; *Fu et al., 2017*; *Greitzer-Antes et al., 2018*). Recent studies have shown that human islet β-cells express both Kv2.1 and Kv2.2 channels, with much higher levels of Kv2.2 mRNA expression (*Blodgett et al., 2015*; *Fu et al., 2017*). *Jensen et al., 2013* reported that Kv2.2 expression is also involved in GSIS. Nevertheless, the functions of the Kv2.2 channel in the in vivo physiology of islet β-cells have not been definitively established.

In this study, we investigated the role of EP1-4 receptors and Kv2.2 in the inhibitory effect of PGE2 on insulin secretion in pancreatic β-cells and *Kcnb2* knockout mice. Our findings indicate that: (1) PGE2 reduces Kv2.2 currents via the EP2/4-PKA signaling pathway; (2) PGE2 inhibits β-cell electrical activity through Kv2.2 channels; (3) knockout of *Kcnb2* abrogates the PGE2-induced inhibition of GSIS. These results suggest that EP2/4 receptors and Kv2.2 channels play crucial regulatory roles in the normal physiological secretion of insulin.

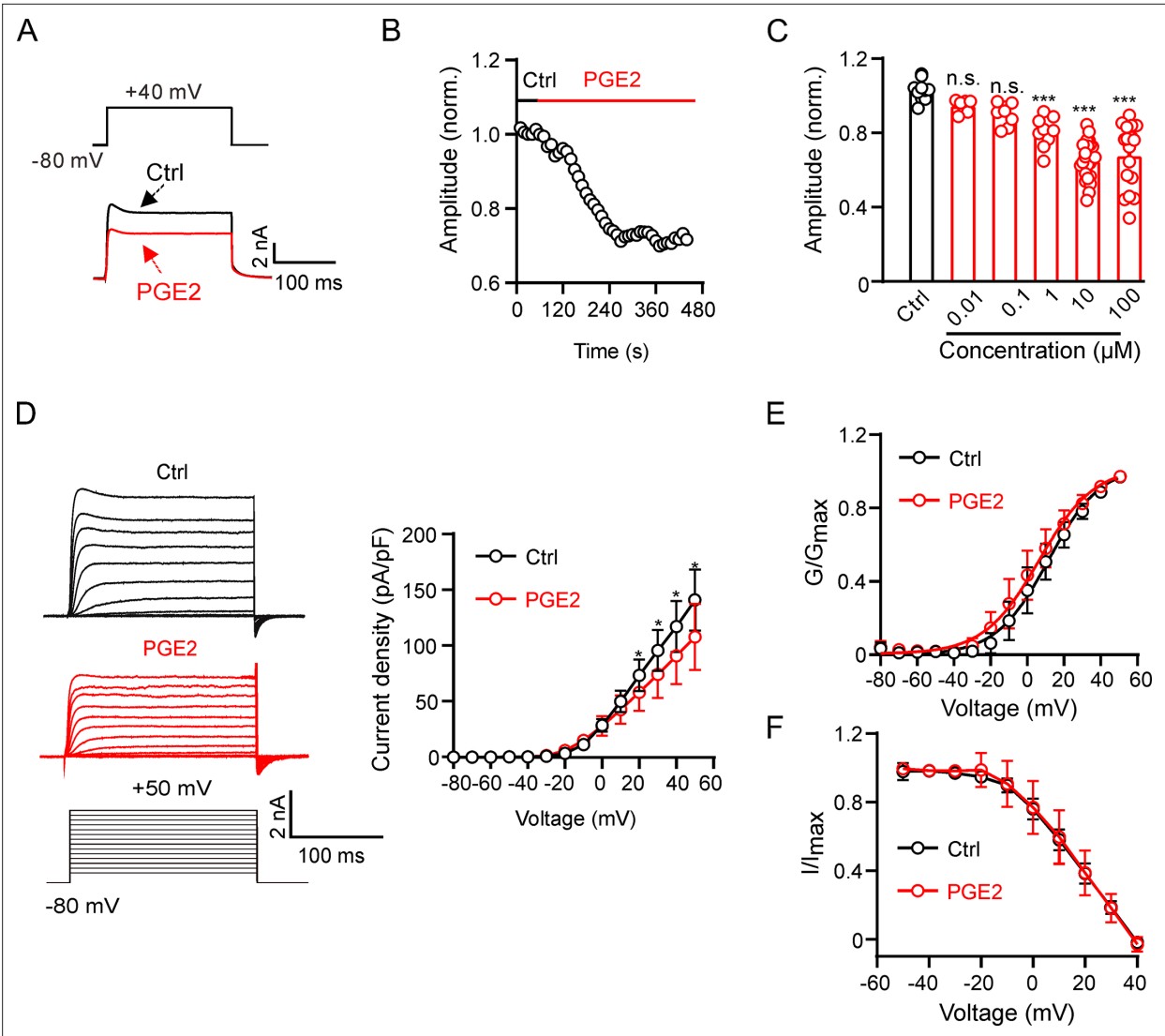

**Figure 1.** Prostaglandin E2 (PGE2) inhibits Kv2.2 channels in HEK293T cells. (**A**) Representative Kv2.2 current traces induced by a depolarization pulse from –80 to +40 mV under the control condition (black) and subsequently in the presence of 10 μM PGE2 (red) in the same HEK293T cell. PGE2 was applied to the extracellular solution and recordings taken at an interval of 10 s. (**B**) The time course of the Kv2.2 current inhibition by 10 μM PGE2. (**C**) PGE2 inhibited Kv2.2 currents in a concentration-dependent manner. n.s., not significant. ***p<0.001. One-way ANOVA with Bonferroni post hoc test (0.01 μM: n=7, p=0.4002; 0.1 μM: n=8, p=0.0672; 1 μM: n=9, ***p=0.0002; 10 μM: n=24, ***p<0.0001; 100 μM: n=17, ***p<0.0001). (**D**) Left, representative Kv2.2 current recordings in response to 200 ms 10 mV depolarizing steps from –80 to +50 mV in the control (top, black) and PGE2-treated (bottom, red) groups. Right, plot of the current-voltage relationship from left (n=7 for each data point). *p<0.05. Two-tailed paired t-test. (**E**) Plot of Kv2.2 current activation curves in the control (black) and PGE2-treated (red, n=7 for each data point) groups. (**F**) Plot of Kv2.2 current inactivation curves in the control (black) and PGE2-treated (red, n=7 for each data point) groups.

The online version of this article includes the following source data for figure 1:

**Source data 1.** Statistical data for *Figure 1B–F*.

## Results

### PGE2 inhibits Kv2.2 currents via the EP2/EP4 signaling pathway in HEK293T cells

We initially conducted experiments to assess the impact of PGE2 on Kv2.2 channels. Kv2.2 channels were overexpressed in HEK293T cells, and Kv2.2 currents were elicited by a 200 ms depolarization pulse from –80 to +40 mV. Extracellular application of 10 μM PGE2 significantly inhibited Kv2.2 currents. This inhibition was rapid, reaching its maximum effect in approximately 6 min (*Figure 1A*

*and B*). The inhibitory effect of PGE2 on Kv2.2 is dose-dependent, with the maximum effect at around 10 μM (*Figure 1C*). Therefore, we used 10 μM PGE2 for the subsequent experiments. The I-V curve demonstrated that PGE2 significantly inhibited Kv2.2 currents at all positive testing potentials above +20 mV (*Figure 1D*). Moreover, 10 μM PGE2 did not alter the steady-state activation and inactivation properties of Kv2.2 currents (*Figure 1E and F*).

Next, we investigated the mechanism of PGE2 inhibition on Kv2.2 channels. PGE2 functions by activating EP1–4 receptors. We investigated the mRNA and protein expression profiles of EP receptors in HEK293T cells. The mRNAs for all four EP receptors were detected in HEK293T cells, with notably higher levels observed for EP2 and EP4 (*Figure 2A*). Furthermore, immunofluorescence results confirmed the presence of protein expression for EP1–4 receptors in HEK293T cells (*Figure 2B*). To determine which EP receptor is responsible for the PGE2-induced inhibition, we applied SC51089 (10 μM, the EP1 receptor antagonist) (*Zhou et al., 2008*), AH6809 (20 μM, the EP2 receptor antagonist) (*Srinivasalu et al., 2020*), L798106 (10 μM, the EP3 receptor antagonist) (*Corboz et al., 2021*), and AH23848 (20 μM, the EP4 receptor antagonist) (*He et al., 2021*) to the extracellular solution to selectively inhibit the respective EP receptors. The acute application of the four antagonists alone had no effect on Kv2.2 currents (*Figure 2C*). Pretreatment with SC51089 and L798106 for 10 min did not alter the PGE2-induced inhibition. However, pretreatment with AH6809 and AH23848 partially blocked the PGE2-induced inhibition (*Figure 2D and E*). Furthermore, the EP2 receptor agonist butaprost (20 μM) and the EP4 receptor agonist CAY10598 (20 μM) both inhibited Kv2.2 currents (*Figure 2F and G*). These results suggest that PGE2 inhibits Kv2.2 via EP2/EP4 signaling pathway.

## PGE2 inhibits Kv2.2 currents via the PKA signaling pathway in HEK293T cells

The EP2 and EP4 receptors couple to Gs, increasing intracellular cAMP formation and activating the PKA signaling pathway (*Sugimoto and Narumiya, 2007*). We investigated whether PGE2 regulates Kv2.2 via PKA. Treatment with 10 μM PGE2 rapidly increased PKA activity in HEK293T cells (*Figure 3A*). The PKA activator, Db-cAMP, could mimic the inhibitory effect of PGE2 on Kv2.2 channels (*Figure 3B*). The PKA inhibitor, Rp-cAMP, had no effect on Kv2.2 currents (*Figure 3C*). However, preincubation with Rp-cAMP blocked the inhibitory effect of PGE2 on Kv2.2 channels in HEK293T cells (*Figure 3D*). These findings illustrate that PGE2 inhibits Kv2.2 currents through the PKA signaling pathway.

We further investigated whether PKA regulates Kv2.2 currents via direct phosphorylation of the channel. The Kv2.2 channel protein contains approximately nine putative phosphorylation sites that conform to the minimal consensus sequence for PKA, according to https://scansite4.mit.edu. The positions of the nine sites are depicted in *Figure 3E*. To identify the specific site responsible for PKA regulation of Kv2.2 channels, we introduced mutations in the amino acid sequence. We replaced all nine amino acids with aspartic acid to mimic the PKA phosphorylation state of Kv2.2 channels. Among the nine mutations, only the S448D mutation yielded a significant reduction in Kv2.2 currents (*Figure 3F*). Notably, the S448D mutation also abrogated the inhibitory effect of PGE2 on Kv2.2 (*Figure 3G*). To further confirm the role of the S448 site in PGE2-induced inhibition of Kv2.2, we also conducted a mutation, changing S448 to alanine (S448A). This mutation effectively prevented the inhibitory effect of PGE2 on Kv2.2 currents (*Figure 3H*). These data indicate that the S448 site is responsible for the PGE2-induced inhibition of Kv2.2 currents.

## PGE2 inhibits endogenous Kv2.2 currents through the EP2/EP4 signaling pathway in INS-1(832/13) β-cells

To investigate the physiological function of PGE2-induced inhibition of Kv2.2, we examined whether PGE2 could also affect the native Kv2.2 currents in pancreatic β-cells. Previous study has demonstrated that Kv2.2 channels contribute to the delayed rectifier outward K$^+$ currents in pancreatic β-cells (*Fu et al., 2017*). Consistent with these studies, we observed the protein expression of Kv2.2 channels in INS-1(832/13) β-cells (*Figure 4A*). The voltage-dependent potassium currents ($I_K$) in INS-1(832/13) cells were elicited by a 200 ms depolarization pulse from –80 to +40 mV, and the extracellular application of 10 μM PGE2 significantly inhibited the $I_K$ (*Figure 4B*).

To assess the contribution of Kv2.2 channels to the PGE2-induced inhibition in INS-1(832/13) cells, we knocked down the expression of Kv2.2 using shRNA. The efficiency of the two shRNAs, named

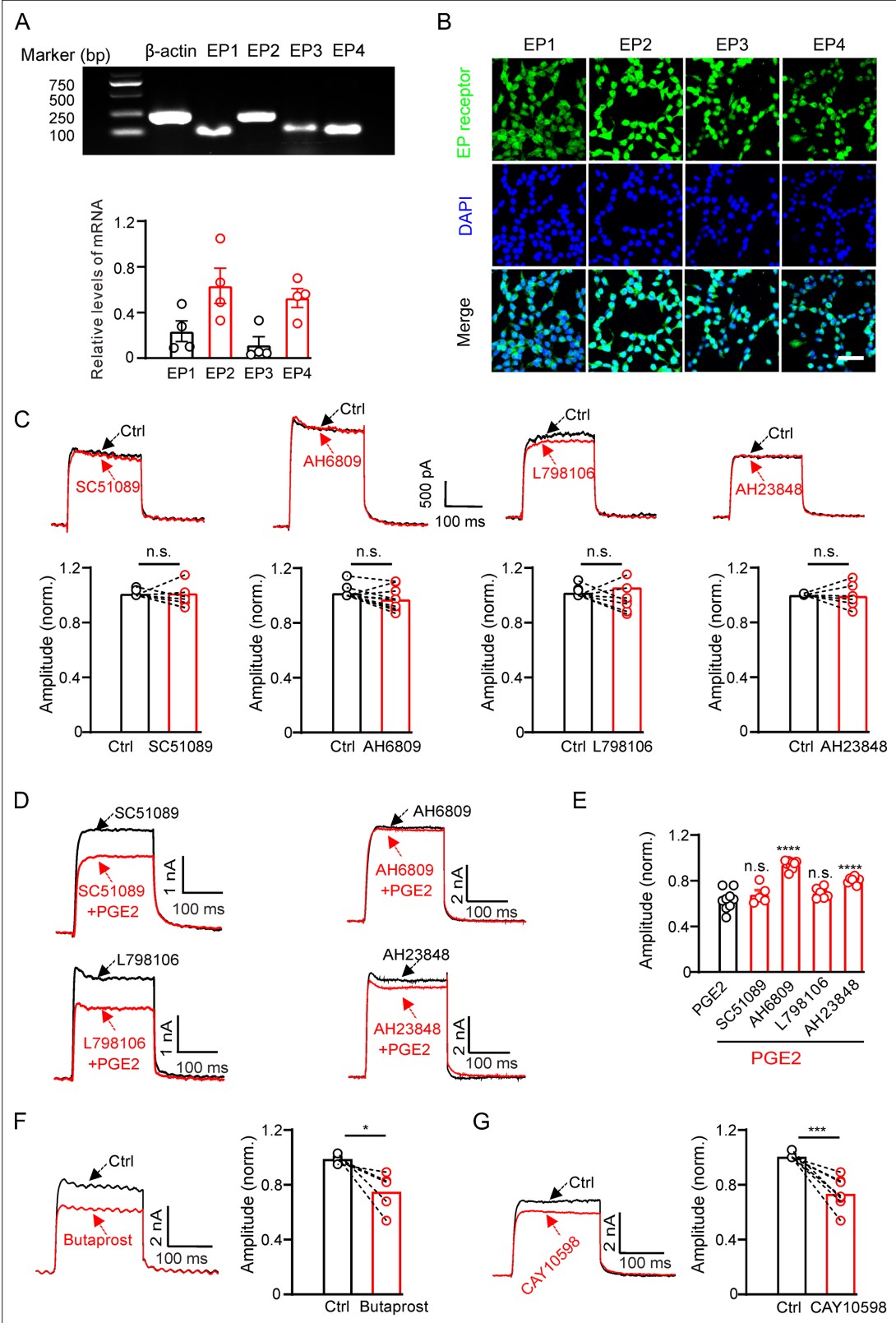

**Figure 2.** Prostaglandin E2 (PGE2) inhibits Kv2.2 currents via the EP2/EP4 signaling pathway in HEK293T cells. (**A**) Top, real-time polymerase chain reaction (RT-PCR) showing the mRNA expression of EP1–4 receptors in HEK293T cells. Bottom, statistics of the mRNA expression of EP1–4 receptors in HEK293T cells (n=4). (**B**) Representative examples of immunofluorescence images showing expression of EP1–4 receptors in HEK293T cells. Scale bar, 20 μm. (**C**) SC51089 (the EP1 receptor antagonist), AH6809 (the EP2 receptor antagonist), L798106 (the EP3 receptor antagonist), and AH23848 (the EP4

*Figure 2 continued on next page*

*Figure 2 continued*

receptor antagonist) per se did not alter Kv2.2 currents. n.s., not significant (SC51089: n=8, p=0.9154; AH6809: n=10, p=0.0661; L798106: n=9, p=0.5581; AH23848: n=6, p=0.8827). (**D**) Representative Kv2.2 current traces induced by a depolarization pulse from –80 to +40 mV in the presence of SC51089, AH6809, L798106, and AH23848 respectively, and subsequently in the presence of an additional 10 µM PGE2 in the same HEK293T cell. (**E**) Statistical analysis showing the effect of EP1–4 antagonists on PGE2-induced inhibition of Kv2.2 channels. ****p<0.001 versus PGE2 alone by a two-tailed unpaired t-test. n.s., not significant (+SC51089: n=5, p=0.3997; +AH6809: n=10, ****p<0.0001; +L798106: n=6, p=0.1785; +AH23848: n=8, ****p<0.0001). (**F**) Left, representative Kv2.2 current traces induced by a depolarization pulse from –80 to +40 mV under the control condition and subsequently in the presence of the EP2 receptor agonist butaprost in the same HEK293T cell. Right, statistics for the amplitude of Kv2.2 currents from left using a two-tailed paired t-test (n=5, *p=0.0306). (**G**) Similar to F, but with the EP4 receptor agonist CAY10598 in the extracellular solution (n=8, ***p=0.0003).

The online version of this article includes the following source data for figure 2:

**Source data 1.** Statistical data for *Figure 2A, C, E, F, and G*.

**Source data 2.** Uncropped DNA gel image for *Figure 2A*.

**Source data 3.** PDF file containing uncropped DNA gel image for *Figure 2A*, indicating the relevant bands and treatments.

KD1-Kv2.2 and KD2-Kv2.2, were first evaluated in HEK293T cells overexpressing Kv2.2. KD2-Kv2.2 showed higher efficiency than KD1-Kv2.2 and was selected for subsequent experiments (*Figure 4C*). Knockdown of *Kcnb2* by KD2-Kv2.2 significantly reduced endogenous K⁺ currents (*Figure 4D*) in INS-1(832/13) cells and abrogated PGE2-induced inhibition of K⁺ currents (*Figure 4E and F*). Furthermore, the EP2 receptor agonist butaprost (20 µM) and the EP4 receptor agonist CAY10598 (20 µM) both inhibited the $I_K$ in INS-1(832/13) cells (*Figure 4G and H*). Pretreatment with TG4155 (1 µM; the EP2 receptor antagonist) (*Jiang et al., 2012*) and GW627368 (10 µM; the EP4 receptor antagonist) (*Evans et al., 2023*) partially blocked the PGE2-induced inhibition, confirming that it involves the same signaling pathway as that in HEK293T cells (*Figure 4I and J*).

## PGE2 reduces electrical activity via Kv2.2 channels in INS-1(832/13) cells

GSIS is associated with β-cell electrical activity, which is regulated by Kv2 potassium channels (*Drews et al., 2010*). Therefore, we tested whether PGE2 affects β-cell electrical activity. We found that 10 µM PGE2 reduced action potential (AP) frequency and increased AP half-width in INS-1(832/13) cells with little effect on the peak amplitude of AP (*Figure 5A and B*). Furthermore, knockdown of *Kcnb2* with Kv2.2-specific shRNA abrogated PGE2-induced effect on INS-1(832/13) cells (*Figure 5C*). This suggests that PGE2 reduces electrical activity via Kv2.2 channels in INS-1(832/13) cells. Since PGE2 inhibits Kv2.2 via EP2/4 receptors, it is plausible to hypothesize that PGE2 reduces electrical activity via EP2/4 receptors in INS-1(832/13) cells. To test the hypothesis, we first investigated the expression of EP receptors in INS-1(832/13) cells. Immunofluorescence results revealed that all EP1–4 receptors are expressed in INS-1(832/13) cells (*Figure 5D*). The specificity of the EP receptor antibody was validated using siRNA knockdown (*Figure 5—figure supplement 1*). In addition, both mouse and human islet β-cells were found to express all four EP receptors (*Figure 5E*). As expected, the EP2 receptor agonist butaprost (20 µM) and the EP4 receptor agonist CAY10598 (20 µM) both inhibited AP frequency in INS-1(832/13) cells (*Figure 6A and B*). This indicates that PGE2 reduces electrical activity via EP2/4 in INS-1(832/13) cells.

## PGE2 regulates insulin secretion through Kv2.2 channels in INS-1(832/13) cells and mice

We next investigated whether Kv2.2 regulates the effect of PGE2 on GSIS in INS-1(832/13) cells. As expected, 10 µM PGE2 significantly reduced GSIS with little effect on basal insulin secretion in INS-1(832/13) cells (*Figure 7A*). Knockdown of *Kcnb2* with KD2 shRNA did not alter basal insulin secretion but reduced GSIS in INS-1(832/13) cells (*Figure 7B*). Moreover, knockdown of *Kcnb2* with KD2 shRNA greatly alleviated the inhibitory effect of PGE2 on GSIS (percent inhibition: scramble, 62%; KD2, 14%, *Figure 7B*). To further validate that PGE2 regulates GSIS through PKA-mediated phosphorylation of the KV2.2 channel, we overexpressed the Kv2.2-S448A mutant channel in INS-1 (832/13) cells. Compared to cells transfected with either an empty vector (as a control) or the wild-type Kv2.2 plasmid, overexpression of Kv2.2-S448A significantly attenuated the inhibitory effect of PGE2 on GSIS (*Figure 7C*).

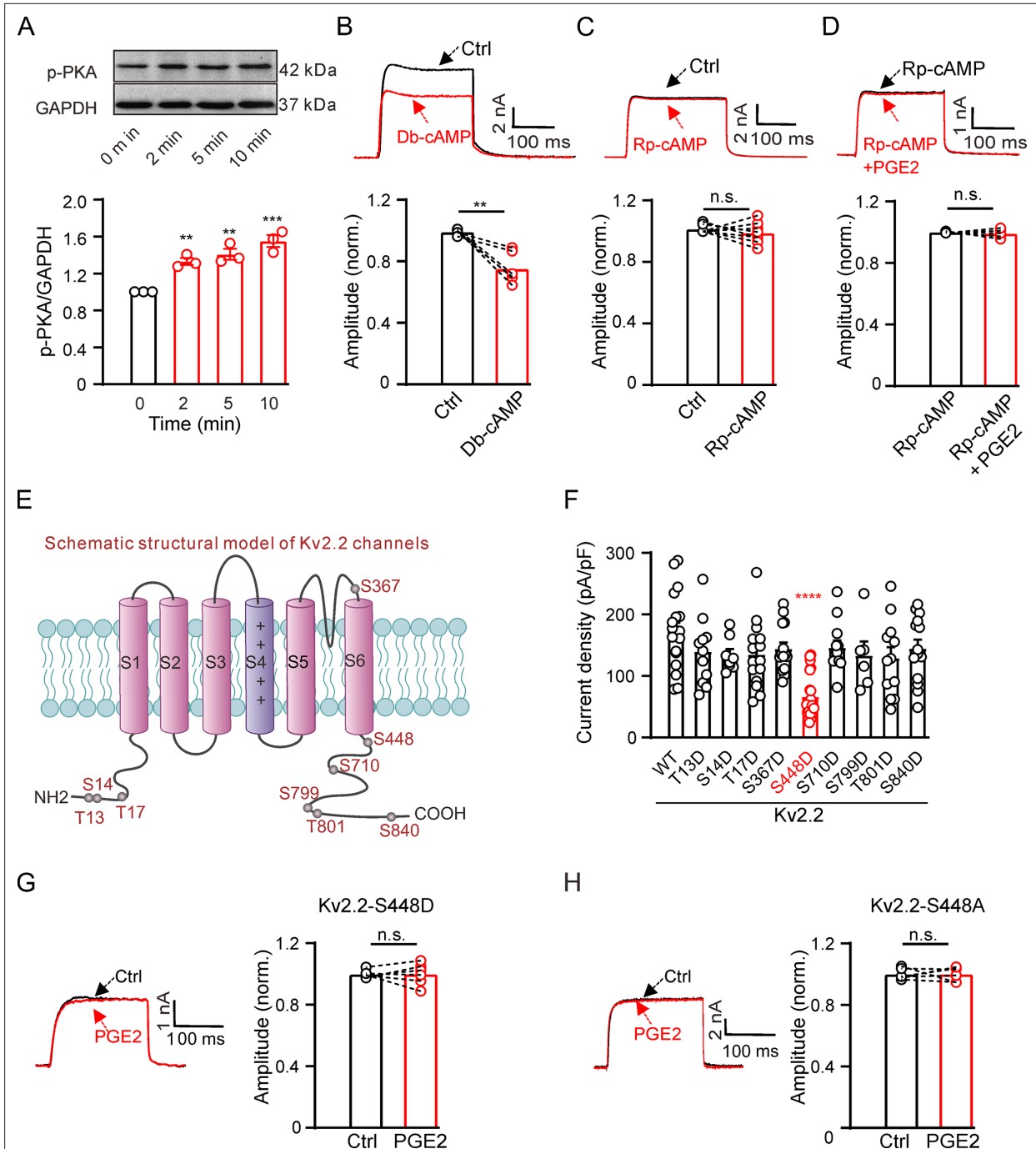

**Figure 3.** Prostaglandin E2 (PGE2) inhibits Kv2.2 currents via the PKA signaling pathway. (**A**) Top, representative western blot showing the PKA phosphorylation level in HEK293T cells following treatment with 10 μM PGE2 for time intervals of 2, 5, and 10 min. Bottom, statistics from three independent experiments using a one-way ANOVA with Bonferroni post hoc test (n=3; 2 min: **p=0.0078; 5 min: **p=0.002; 10 min: ***p=0.0002). (**B**) Top, representative Kv2.2 current traces induced by a depolarization pulse from –80 to +40 mV under the control condition (black) and subsequently in the presence of 10 μM Db-cAMP (red) in the same HEK293T cell. Bottom, statistics for the amplitude of Kv2.2 currents from top using a two-tailed paired t-test (n=6, **p=0.0035). (**C**) Similar to B, but with Rp-cAMP in the extracellular solution (n=8, p=0.3851). (**D**) Top, representative Kv2.2 current traces induced by a depolarization pulse from –80 to +40 mV in the presence of 10 μM Rp-cAMP (black) and subsequently in the presence of an additional 10 μM PGE2 (red) in the same HEK293T cell. Bottom, statistics for the amplitude of Kv2.2 current from top using a two-tailed paired t-test (n=5, p=0.6425). n.s., not significant. (**E**) Schematic structural models of Kv2.2 channels indicate the positions of nine potential PKA phosphorylation sites. (**F**) Statistics for the amplitude of wild-type and various mutant Kv2.2 channel currents, induced by a depolarization pulse from –80 to +40 mV (n=6–24). ****p<0.0001 compared to wild-type Kv2.2 by a one-way ANOVA with Bonferroni post hoc test. (**G**) Left, representative Kv2.2-S448D mutant channel

*Figure 3 continued on next page*

*Figure 3 continued*

current traces induced by a depolarization pulse from –80 to +40 mV under the control condition (black) and subsequently in the presence of 10 µM PGE2 (red) in the same HEK293T cell. Right, statistics for the amplitude of Kv2.2 current from left using a two-tailed paired t-test (n=6, p>0.9999). n.s., not significant. (**H**) Similar to G, this section presents data for the Kv2.2-S448A mutant channel in HEK293T cells (n=6, p>0.9999).

The online version of this article includes the following source data for figure 3:

**Source data 1.** Statistical data for *Figure 3A, B, C, D, F, G, and H*.

**Source data 2.** Uncropped western blot images for *Figure 3A*.

**Source data 3.** PDF file containing uncropped western blot images for *Figure 3A*, indicating the relevant bands and treatments.

To investigate the physiological role of Kv2.2 during GSIS in vivo, a mouse model was utilized in which the Kv2.2 coding gene (*Kcnb2*) is disrupted at exon 2 (*Figure 7—figure supplement 1A*). The targeting cassette removes 446 bp in exon 2 of the *Kcnb2* gene sequence. The disrupted sequence is detected using PCR to produce one amplicon with one primer inside the targeting sequence in combination with one *Kcnb2*-specific primer outside the targeting sequence (*Figure 7—figure supplement 1A*). The Kv2.2 protein was detected only in wild-type islets using a Kv2.2-specific anti-body (*Figure 7—figure supplement 1B*). The body weight of the *Kcnb2* knockout mice was similar to that of the wild-type littermates (*Figure 7—figure supplement 1C*). *Kcnb2*$^{-/-}$ mice were assessed for possible impairment in glucose homeostasis using an intraperitoneal glucose tolerance test. *Kcnb2*$^{-/-}$ mice had similar fasting blood glucose levels with control group (*Figure 7D*). PGE2 treatment wors-ened glucose tolerance in control animals but had little effect on *Kcnb2*$^{-/-}$ mice (*Figure 7D*). This suggests that PGE2 modulates GSIS in vivo through Kv2.2 channels. The *Kcnb2*$^{-/-}$ gene knockout mice used in this study are global knockouts. To further confirm the role of Kv2.2 in insulin secretion, we isolated pancreatic islets for GSIS experiments. Islets from *Kcnb2*$^{-/-}$ mice exhibited reduced GSIS compared to islets from wild-type mice (*Figure 7E*). While PGE2 reduced GSIS in islets from wild-type mice, it had little effect on GSIS in islets from *Kcnb2*$^{-/-}$ mice (*Figure 7E*).

## PGE2 reduces electrical activity via Kv2.2 channels in mouse β-cells within islets

Finally, we investigated the effect of PGE2 on the electrical activity of β-cells within islets. The $I_k$ in *Kcnb2*$^{-/-}$ β-cells showed significantly reduced amplitudes (*Figure 8A*). Similar to our observations in INS-1(832/13) cells above, 10 µM PGE2 inhibited $I_k$ in mouse β-cells, and Kv2.2 ablation abol-ished the inhibitory effect of PGE2 (*Figure 8B*). PGE2 reduced the firing frequency of AP induced by 20 mM glucose in β-cells within islets, while had little effect on the AP peak amplitude and half-width (*Figure 8C*). Furthermore, the effect of PGE2 on mouse islet β-cell electrical activity was abrogated by knockout of *Kcnb2* (*Figure 8D*).

## Discussion

While EP3 has been demonstrated to exert an inhibitory influence on GSIS, increasing evidence suggests that the impact of EP3 may manifest primarily when β-cell dysfunction is already established, as observed in T2DM (*Carboneau et al., 2017*; *Truchan et al., 2021*). In our current investigation, we unveil that Kv2.2 channels play a crucial role in regulating β-cell repetitive firing and insulin secretion. PGE2 inhibits Kv2.2 channels via the EP2/4 signaling pathway, consequently impeding GSIS in normal β-cells.

Even though the previous study reported that the EP2 protein-encoding gene *Ptger2* was not identified by RNA-seq in mouse islets (*Ku et al., 2012*), other studies have shown that the mRNA expression of all four EPs is observed in rat, mouse, and human pancreatic islets (*Bramswig et al., 2013*; *Tran et al., 2002*; *Vennemann et al., 2012*). In our current study, we provide evidence demonstrating the presence of protein expression for all four types of PGE2 receptors in both human and mouse pancreatic β-cells. In comparison to EP3, our knowledge of the impact of EP1, EP2, and EP4 on insulin secretion is currently limited. Tran et al. have reported that the EP1 antag-onist fails to impede the inhibitory effects of IL-1β on GSIS in isolated rat islets. EP3 agonists, such as misoprostol or sulprostone, decrease GSIS through Gi proteins in rat islets (*Tran et al., 2002*). A recent study has unveiled that the activation of Gi/o protein-coupled receptors leads to the

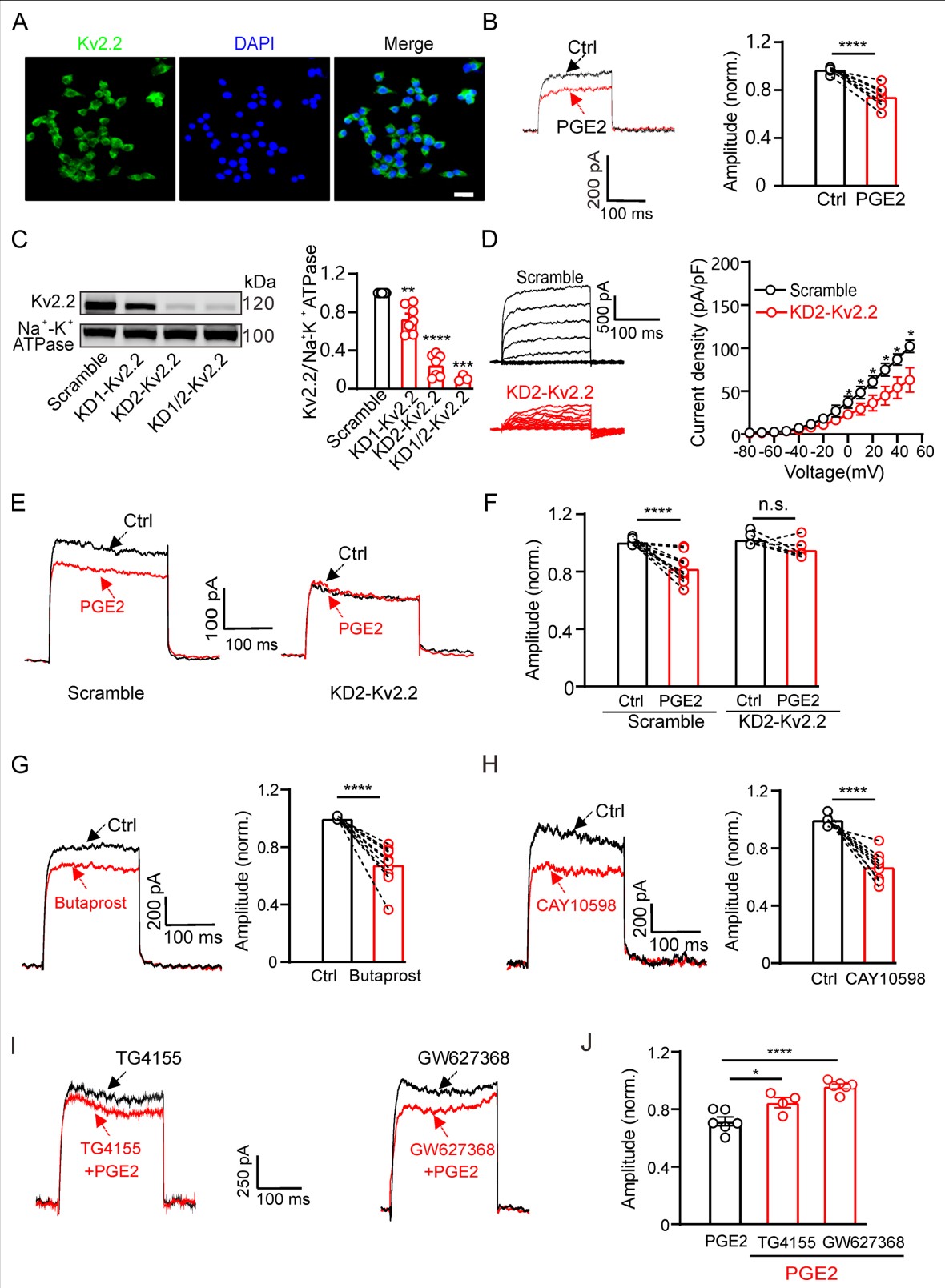

**Figure 4.** Prostaglandin E2 (PGE2) inhibits native Kv2.2 channels in INS-1(832/13) cells. (**A**) Representative examples of immunofluorescence images showing high expression of Kv2.2 channels in INS-1(832/13) cells. Scale bar, 20 μm. (**B**) Left, representative $I_K$ traces induced by a depolarization pulse from –80 to +40 mV under the control condition (black) and subsequently in the presence of 10 μM PGE2 (red) in the same INS-1(832/13) cell. Right, statistics for the amplitude of $I_K$ from left using a two-tailed paired t-test (n=10, ****p<0.0001). (**C**) Left, representative examples of western blot images

*Figure 4 continued on next page*

*Figure 4 continued*

showing the effects of the two shRNA oligos (KD1-Kv2.2 and KD2-Kv2.2) on Kv2.2 channel surface expression in HEK293T cells transfected with Kv2.2. Right, statistics for cell surface expression of Kv2.2 channels from left using a two-tailed unpaired t-test. KD1-Kv2.2: n=7, **p=0.0066; KD2-Kv2.2: n=7, ****p<0.0001; KD1/KD2-Kv2.2: n=3, ***p=0.0005, compared with scramble. (**D**) Knockdown of *Kcnb2* significantly reduced $I_k$ amplitude in INS-1(832/13) cells (n=6, *p<0.05). (**E**) Left, representative $I_k$ traces induced by a depolarization pulse from –80 to +40 mV under the control condition (black) and subsequently in the presence of 10 µM PGE2 (red) in the same INS-1(832/13) cell transfected with scramble control or KD2-Kv2.2. (**F**) Knockdown of *Kcnb2* abrogated the inhibitory effect of PGE2 on $I_k$ in INS-1(832/13) cells. Scramble-PGE2: n=11, ****p<0.0001; KD2-PGE2: n=6, p=0.1227; two-tailed paired t-test. (**G**) Left, representative Kv2.2 current traces induced by a depolarization pulse from –80 to +40 mV under the control condition and subsequently in the presence of the EP2 receptor agonist butaprost in the same INS-1(832/13) cell. Right, statistics for the amplitude of Kv2.2 currents from left using a two-tailed paired t-test (n=10, ****p<0.0001). (**H**) Similar to G, but with the EP4 receptor agonist CAY10598 in the extracellular solution (n=9, ****p<0.0001). (**I**) Representative Kv2.2 current traces induced by a depolarization pulse from –80 to +40 mV in the presence of TG4155 or GW627368, followed by the addition of 10 µM PGE2 in the same INS-1(832/13) cell. (**J**) Statistical analysis showing the effects of EP2 and EP4 antagonists on PGE2-induced inhibition of Kv2.2 channels. *p<0.05 and ****p<0.0001 versus PGE2 alone by a two-tailed unpaired t-test (+TG4155: n=4, *p=0.0242; +GW627368: n=6, ****p<0.0001).

The online version of this article includes the following source data for figure 4:

**Source data 1.** Statistical data for *Figure 4B, C, D, F, G, H, and J*.

**Source data 2.** Uncropped western blot images for *Figure 4C*.

**Source data 3.** PDF file containing uncropped western blot images for *Figure 4C*, indicating the relevant bands and treatments.

stimulation of Na$^+$/K$^+$ ATPase. This stimulation, in turn, hyperpolarizes the membrane potential of β-cells, consequently suppressing β-cell electrical excitability and insulin secretion (*Dickerson et al., 2022*). Consequently, one would anticipate that PGE2 activates Gi protein-coupled EP3 receptors, thereby inhibiting insulin secretion through membrane potential hyperpolarization and the reduction in β-cell electrical excitability. Surprisingly, no alterations in β-cell membrane potential were observed following PGE2 treatments. We found that PGE2 inhibits Kv2.2 channels and electroactivity of β-cells via G$_s$-coupled EP2/4 receptors instead of EP3 receptors. PGE2 is likely to predominantly exert its effects via the EP2/4 receptors in normal β-cells. However, in instances of established β-cell dysfunction, such as in T2DM, it appears that PGE2 may act through the upregulated EP3 receptors (*Kimple et al., 2013*). This study found that activating either EP2 or EP4 receptors can inhibit Kv2.2 channels, and further research is needed to determine which receptor is the primary regulatory factor for GSIS.

Both Kv2.1 and Kv2.2 contribute to the delayed outward K$^+$ current in human β-cells, and their mRNA expression in diabetic islets is lower than that in nondiabetic islets (*Fu et al., 2017*). Kv2.1 channels in cell membrane exist in two forms: clustered and non-clustered. Non-clustered Kv2.1 channels conduct K$^+$ normally, while clustered Kv2.1 channels are barely conductive (*Fox et al., 2013*). A recent study has shown that Kv2.1 channels form clusters in INS-1(832/13) cells and human β-cells (*Fu et al., 2017*). Kv2.1-mediated insulin exocytosis requires Kv2.1 clustering and a direct interaction with syntaxin 1A but is not dependent on its electrical function (*Dai et al., 2012*; *Fu et al., 2017*). Consistent with the findings of the previous study (*Jensen et al., 2013*), we observed a significant reduction in GSIS in INS-1(832/13) cells upon knockdown of *Kcnb2*. Moreover, knockout of *Kcnb2* also reduced GSIS in mouse islets. Kv2.2 channels play a crucial role in maintaining repetitive AP firing by promoting the recovery of voltage-gated sodium channels from inactivation (*Johnston et al., 2008*). Inhibition of Kv2.2 channels significantly reduces the repetitive firing in medial nucleus of the trapezoid body neurons and cortical pyramidal neurons (*Johnston et al., 2008*; *Wang et al., 2024*). In the present study, we found that the major effect of PGE2-induced inhibition of Kv2.2 is to reduce the firing rate in β-cells, suggesting that Kv2.2 plays a key role in the β-cell repetitive firing upon stimulation. The PGE2-induced reduction in β-cell repetitive firing would decrease calcium channel opening, and, thus, insulin secretion. Importantly, our findings pertain to physiological conditions, and while we demonstrate the inhibitory effects of PGE2 on Kv2.2 channels in normal β-cells, the role of this pathway under diabetic conditions requires further investigation and will be the focus of future studies.

In summary, this study uncovers a previously unknown role of the EP2/4-PKA-Kv2.2 signaling pathway in the inhibitory effect of PGE2 on GSIS in normal β-cells. This provides valuable insights into the complex interplay between prostaglandins, potassium channels, and insulin secretion, contributing to our understanding of pancreatic islet function and potential implications for diabetes mellitus.

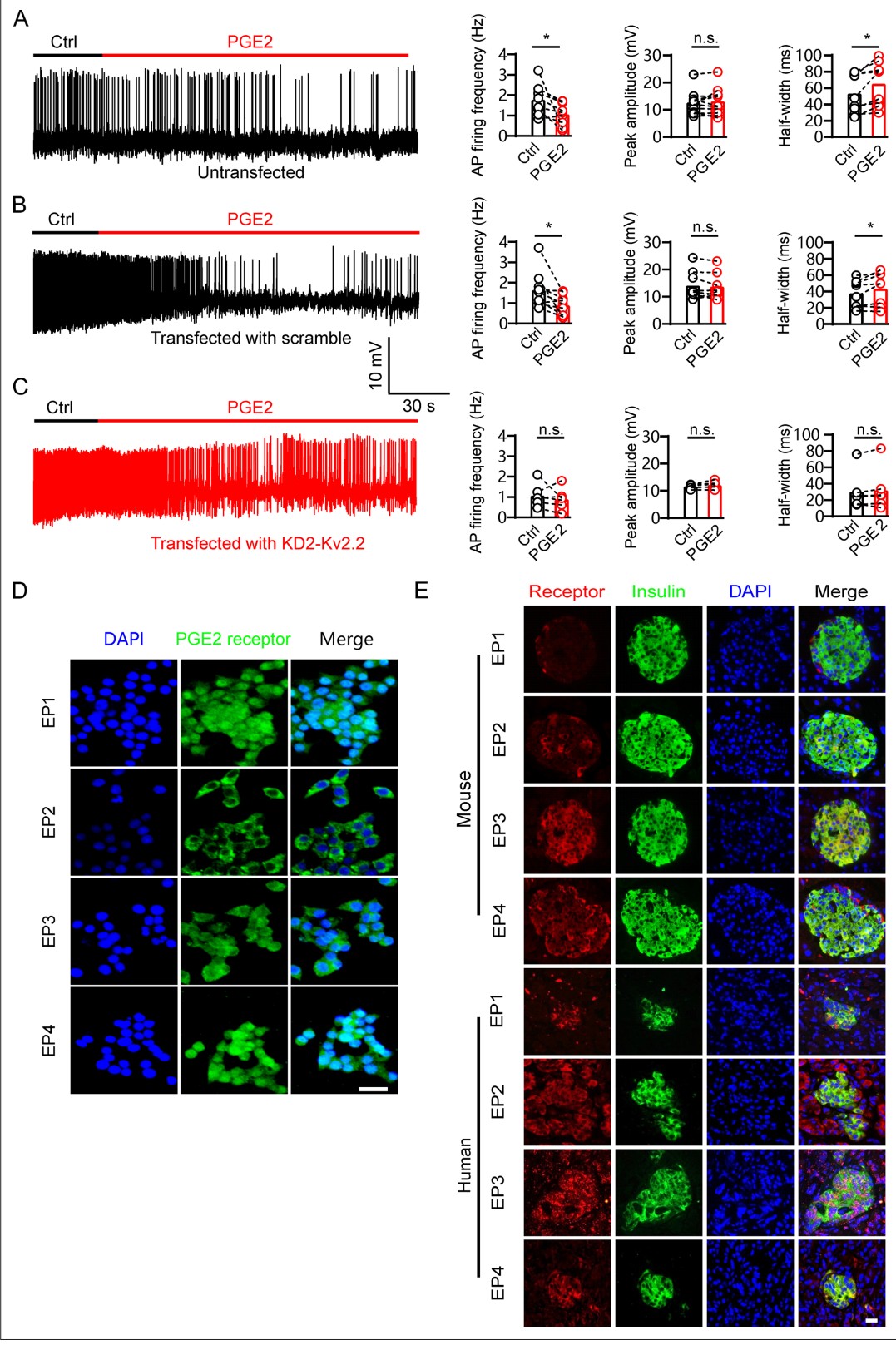

**Figure 5.** Prostaglandin E2 (PGE2) reduces β-cell electrical excitability through Kv2.2 channels. (**A**) Left, representative action potential (AP) firings induced by 20 mM glucose under the control condition and subsequently in the presence of 10 μM PGE2 in the same INS-1(832/13) cell. Right, statistics for the AP firing frequency (*p=0.0385), amplitude (p=0.9478), and half-width (*p=0.0101) from A (n=9). Two-tailed paired t-test.

*Figure 5 continued on next page*

*Figure 5 continued*

(**B**) Similar to A, but the INS-1(832/13) cells were transfected with scramble-Kv2.2 (n=8) (frequency: *p=0.0285; amplitude: p=0.6603; half-width: *p=0.0281). (**C**) Similar to B, but the INS-1(832/13) cells were transfected with KD2-Kv2.2 (n=6). n.s., not significant (frequency: p=0.4564; amplitude: p=0.1601; half-width: p=0.3034). (**D**) Representative immunofluorescence images showing expression of EP1–4 receptors in INS-1(832/13) cells. Scale bar, 20 µm. (**E**) Representative immunofluorescence images showing expression of EP1–4 receptors in mouse and human islets. Scale bar, 20 µm.

The online version of this article includes the following source data and figure supplement(s) for figure 5:

**Source data 1.** Statistical data for *Figure 5A, B, and C*.

**Figure supplement 1.** Validation of EP receptor antibody specificity in INS-1(83 2/13) cells using siRNA knockdown.

**Figure supplement 1—source data 1.** Statistical data for *Figure 5—figure supplement 1*.

**Figure supplement 1—source data 2.** Uncropped western blot images for *Figure 5—figure supplement 1*.

**Figure supplement 1—source data 3.** PDF file containing uncropped western blot images for *Figure 5—figure supplement 1*, indicating the relevant bands and treatments.

# Materials and methods

## Cell culture

Human embryonic kidney (HEK293T) cells were purchased from the cell bank of the Chinese Academy of Science. HEK293T cells were cultured in DMEM supplemented with 10% fetal bovine serum and 1% penicillin-streptomycin solution. INS-1(832/13) β-cells (*Soltani et al., 2011*) were cultured in RPMI 1640 with 10% fetal bovine serum, 1% penicillin-streptomycin solution, and 0.1% β-mercaptoethanol. All the reagents were purchased from Thermo Fisher Scientific (Waltham, MA, USA).

## Molecular biology

Plasmids for rat Kv2.2 (NM_054000.2) channels in pEGFPN1 vectors were as previously reported (*Li et al., 2022*). T13D, S14D, T17D, S367D, S448D, S710D, S799D, T801D, S840D, and S448A mutations of the Kv2.2 channel were achieved by PCR-based site-directed mutagenesis using ClonExpress Multis One Step Cloning Kit (Vazyme, Jiangsu, China). All mutations were confirmed by sequencing. HEK293T cells were transiently transfected with wild-type or mutant Kv2.2 channels for 24 hr using Lipofectamine 2000 (Thermo Fisher Scientific, Waltham, MA, USA) before patch clamp recordings. For knockdown plasmids targeting Kv2.2, the shRNA hairpin sequences were inserted into BamHI and HindIII sites of the PAAV-shRNA targeting vector. Oligonucleotides specifying the shRNA are 5'-GGAGCAGATGAACGAAGAACT-3'(KD1-Kv2.2), 5'-GCTGGAGATGCTATACAATGA-3' (KD2-Kv2.2), and 5'-GCACCCAGTCCGCCCTGAGCAAA-3' (scramble). Total RNA extraction and quantitative real-time polymerase chain reaction (qRT-PCR) were performed as previously described (*Yang et al., 2016*). Briefly, qRT-PCR was performed in 20 µL reactions containing: 2 µL of template, 0.4 µmol/L of each paired primer, and SYBR Green PCR master mix. The thermo-cycling conditions were 94°C, 10 min; 40 cycles of 95°C, 30 s; 55°C, 30 s; 72°C, 60 s; and 72°C, 8 min. Results were normalized by β-actin mRNA. Primers were as previously reported (*Hoshino et al., 2007*): EP1 (PTGER1, forward, reverse): 5'- accttctttggcggctct and 5'- GCACGACACCACCATGATAC; EP2 (PTGER2): 5'- CCACCTCA TTCTCCTGGCTA and 5'- CGACAACAGAGGACTGAACG; EP3 (PTGER3): 5'- AGCTTATGGGGATCAT GTGC and 5'- TCTGCTTCTCCGTGTGTGTC; EP4(PTGER4): 5'-TGCGAGTATTCGTCAACCAG and 5'-GGTCTAGGATGGGGTTCACA; β-actin: 5' -GGACTTCGAGCAAGAGATGG and 5'-AGCACTGTGTTG GCGTACAG.

## Western blot

HEK293T cells were treated with PGE2 for 2, 5, and 10 min, respectively. HEK293T cell homogenates were prepared using a lysis buffer (20 mM HEPES, 0.5% NP-40, 150 mM NaCl, 10% glycerol, 2 mM EDTA, 50 mM NaF, 0.1 mM $Na_3VO_4$) with protease inhibitor (P8340, Sigma, St. Louis, MO, USA) and phosphatase inhibitor (P5726, Sigma, St. Louis, MO, USA) cocktail. The protein samples were separated by 10% SDS-PAGE and then transferred to polyvinylidene fluoride membranes (1620177, Bio-Rad, Hercules, CA, USA). The membranes were blocked with 10% nonfat dry milk in Tris-buffered saline with Tween-20 for 1 hr at room temperature and then incubated with primary antibodies

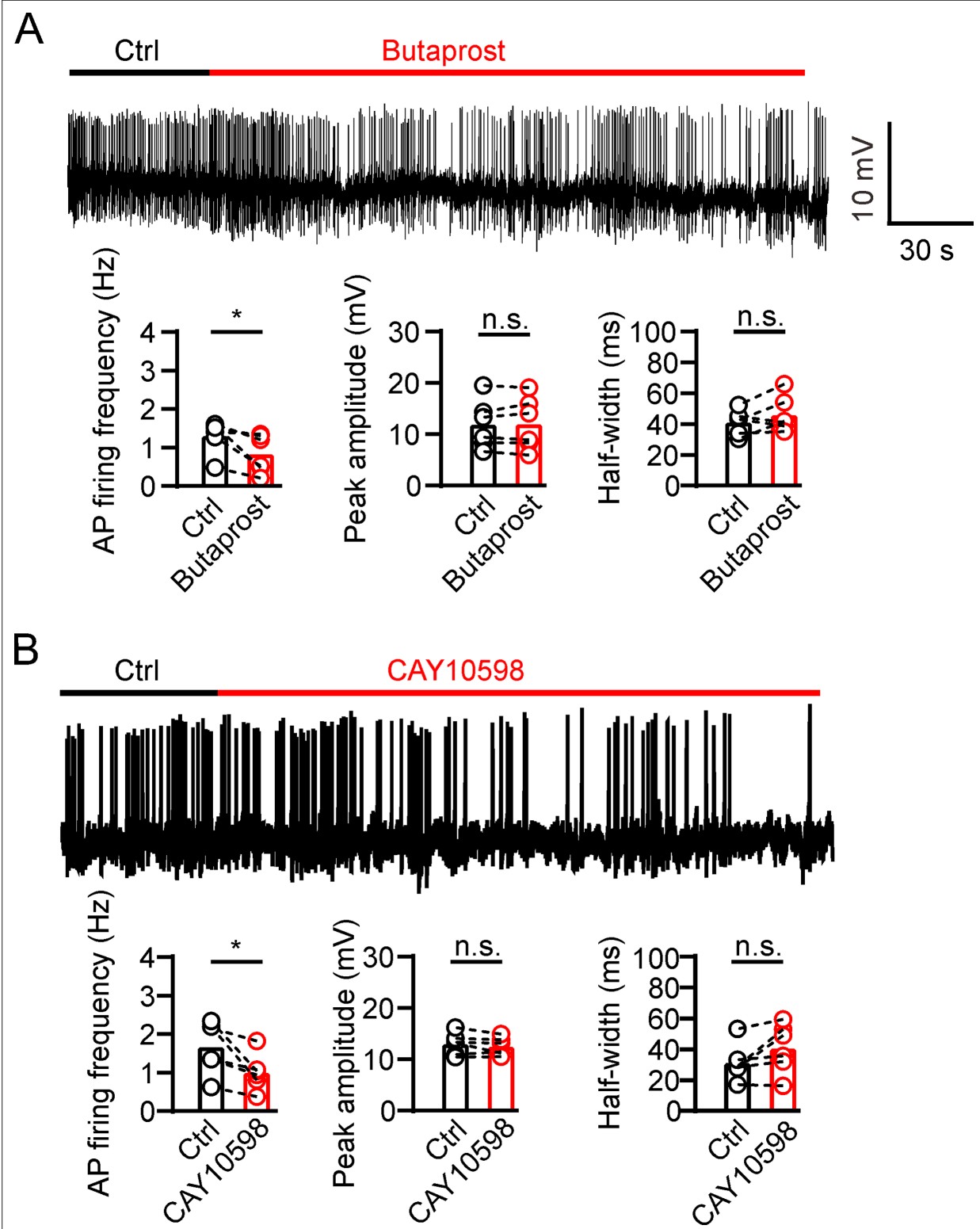

**Figure 6.** Prostaglandin E2 (PGE2) reduces β-cell electrical excitability through EP2/4. (**A**) Top, representative action potential (AP) firings induced by 20 mM glucose under the control condition and subsequently in the presence of 20 μM butaprost in the same INS-1(832/13) cell. Bottom, statistics for the AP firing frequency (*p=0.0312), amplitude (p=0.7354), and half-width (p=0.2067) from top (n=6). Two-tailed paired t-test. (**B**) Similar to A, but in the presence of 20 μM CAY10598 (n=6), AP firing frequency (*p=0.0291), amplitude (p=0.2211), half-width (p=0.0753).

The online version of this article includes the following source data for figure 6:

**Source data 1.** Statistical data for *Figure 6A and B*.

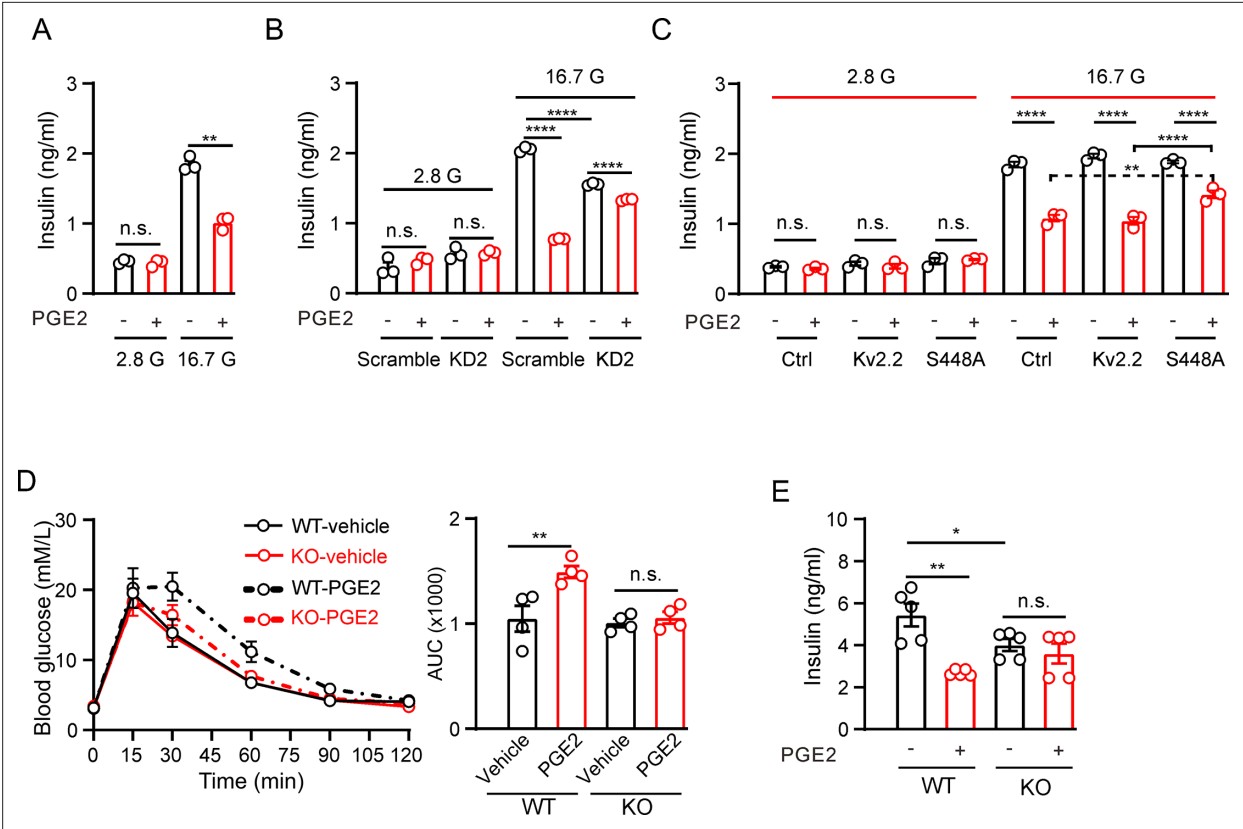

**Figure 7.** Prostaglandin E2 (PGE2) inhibits glucose-stimulated insulin secretion (GSIS) through Kv2.2 channels. (**A**) Effects of PGE2 on insulin secretion in INS-1(832/13) cells under basal (2.8 mM, 2.8 G) or stimulatory (16.7 mM, 16.7 G) glucose concentrations (n=3). 2.8 G, PGE2: n.s., not significant (p=0.9497); 16.7 G, PGE2: **p=0.0082. Two-tailed unpaired t-test. (**B**) Knockdown of *Kcnb2* reduced GIGS and greatly alleviated the PGE2-induced inhibition of GSIS in INS-1(832/13) cells (n=3). ****p<0.0001; n.s., not significant. Two-tailed unpaired t-tests. (**C**) Effects of PGE2 on insulin secretion in INS-1(832/13) cells transfected with empty vectors, Kv2.2, or Kv2.2-S448A mutant channels under basal (2.8 mM, 2.8 G) or stimulatory (16.7 mM, 16.7 G) glucose concentrations (n=3). Overexpression of Kv2.2-S448A mutant channels greatly alleviated the PGE2-induced inhibition of GSIS in INS-1(832/13) cells. n.s., not significant; ****p<0.0001; **p=0.0096. Two-tailed unpaired t-test. (**D**) Left, the effect of PGE2 on the glucose tolerance test in *Kcnb2$^{-/-}$* and control animals (n=4 animals per group). Right, statistics for AUC from left. **p=0.0064. n.s., not significant (p=0.9684). Two-tailed unpaired t-tests. (**E**) Statistics for the effect of PGE2 on GSIS (16.7 mM glucose) in isolated islets from *Kcnb2$^{-/-}$* and control animals. N=5 animals per group. *p=0.0483, **p=0.001. n.s., not significant (p=0.4766). Two-tailed unpaired t-tests.

The online version of this article includes the following source data and figure supplement(s) for figure 7:

**Source data 1.** Statistical data for *Figure 7A–E*.

**Figure supplement 1.** Generation of *Kcnb2* knockout mice.

**Figure supplement 1—source data 1.** Statistical data for *Figure 7—figure supplement 1C*.

**Figure supplement 1—source data 2.** Uncropped DNA gel image for *Figure 7—figure supplement 1A*.

**Figure supplement 1—source data 3.** PDF file containing uncropped DNA gel image for *Figure 7—figure supplement 1A*, indicating the relevant bands and treatments.

overnight at 4°C (anti-phospho-PKA, 1:1000, 5661, Cell Signaling Technology, Danvers, MA, USA; anti-GAPDH, 1:1000; AG019, Beyotime, Shanghai, China). The blots were developed using enhanced chemiluminescence reagents and imaged using the ChemiDoc XRS⁺ imaging system from Bio-Rad (Hercules, CA, USA) and the manufacturer's software.

## Immunofluorescence

HEK293T and INS-1(832/13) cells were fixed in 4% paraformaldehyde (PFA) for 15 min, washed, and blocked (10% donkey serum, 1% BSA) for 2 hr at room temperature. Cells were then incubated in primary antibody solution (primary antibody [anti-EP1: 1:200, Ab217925, Abcam, Cambridge, UK; anti-EP2: 1:200, Ab167171, Abcam, Cambridge, UK; anti-EP3: 1:200, SC-57105, Santa Cruz

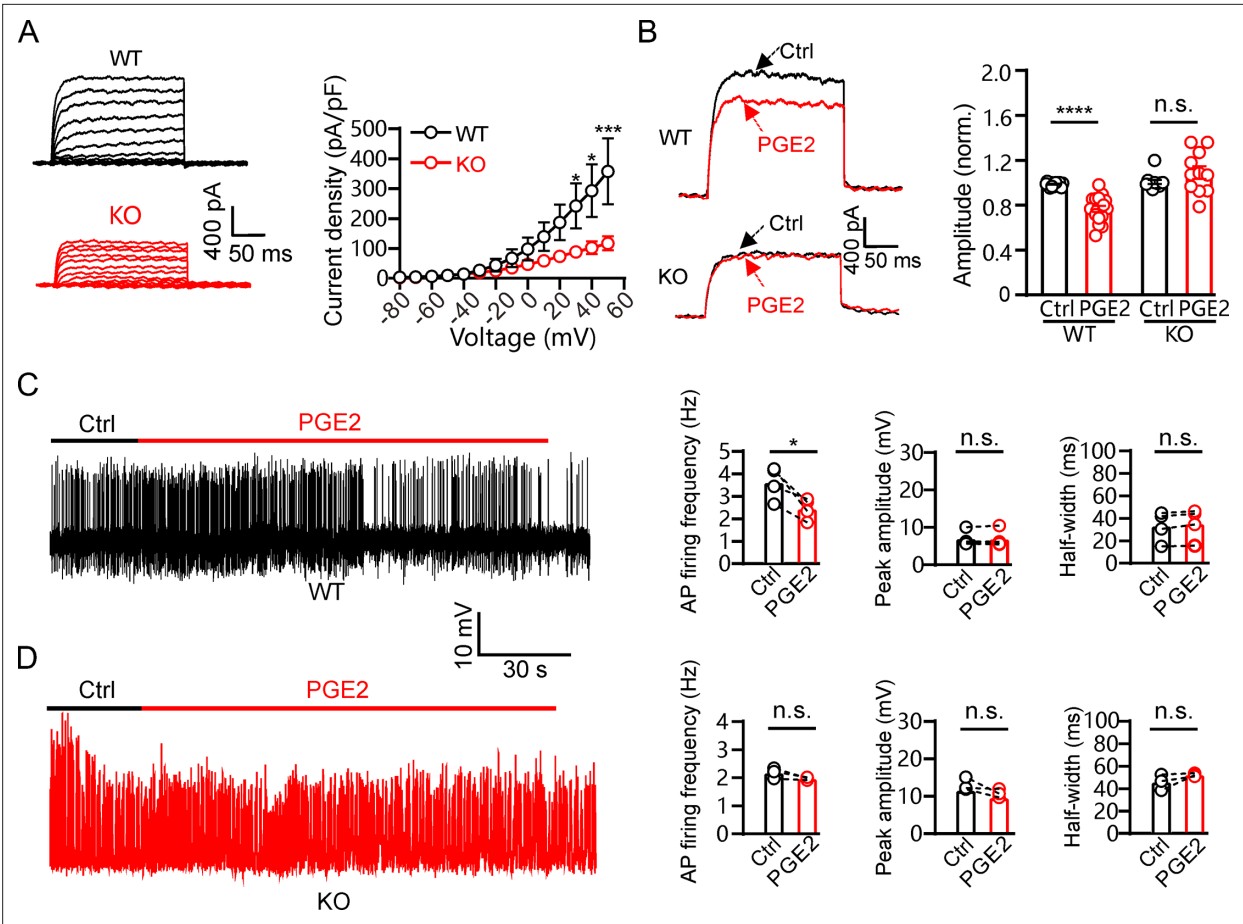

**Figure 8.** Prostaglandin E2 (PGE2) inhibits mouse pancreatic β-cell electrical excitability through Kv2.2 channels. (**A**) Left, representative $I_K$ recordings in response to 200 ms depolarization pulses from – 80 to +50 mV in wild-type and $Kcnb2^{-/-}$ knockout mouse pancreatic β-cells. Right, plot of the current-voltage relationship from left (n=7 for each data point). (**B**) Left, representative $I_K$ traces induced by a depolarization pulse from –80 to +40 mV under the control condition and subsequently in the presence of 10 µM PGE2 in the same β-cell from wild-type or $Kcnb2^{-/-}$ mice. Right, statistics for the $I_K$ amplitude using a two-tailed paired t-test (n=17). ****p<0.0001; n.s., not significant (n=12, p=0.6317). (**C**) Left, representative action potential (AP) firings induced by 20 mM glucose under the control condition and subsequently in the presence of 10 µM PGE2 in the same β-cell within wild-type mouse pancreatic islets. Right, statistics for the AP firing frequency (*p=0.0288), amplitude (p=0.8589), and half-width (p=0.0563) from left (n=4). Two-tailed paired t-tests. (**D**) Left, representative AP firings induced by 20 mM glucose under the control condition and subsequently in the presence of 10 µM PGE2 in the same β-cell within $Kcnb2^{-/-}$ mouse pancreatic islets. Right, statistics for the AP firing frequency (p=0.1536), amplitude (p=0.1981), and half-width (p=0.2385) from left (n=3). Two-tailed paired t-tests.

The online version of this article includes the following source data for figure 8:

**Source data 1.** Statistical data for *Figure 8A–D*.

Biotechnology, Santa Cruz, USA; anti-EP4: 1:200, Ab217966, Abcam, Cambridge, UK; anti-Kv2.2: 1:200, APC-120, Alomone Labs, Jerusalem, IL, USA], 1% horse serum and 0.3% phosphate buffered saline with Tween 20 [PBST]) for 1 day at 4°C. Cells were washed three times with PBST and incubated overnight at 4°C in a secondary antibody solution containing Cy3-labeled goat anti-rat IgG (1:500, A0516, Beyotime, Shanghai, China) or Cy3/FITC-labeled goat anti-mouse IgG (1:500, A0521, Beyotime, Shanghai, China), 1% horse serum, and 0.3% PBST. Following another wash with PBST, the cells were treated with DAPI and imaged using the Nikon A1⁺ Confocal Microscope System. For mouse and human pancreatic tissue, after overnight treatment in 4% PFA, the tissues were washed three times with PBS and then dehydrated in a fresh 30% sucrose solution. Subsequently, the pancreatic tissue was embedded in optimal cutting temperature compound and sliced into 25 µm sections using a cryostat. Slices were blocked with 10% horse serum and 0.3% PBST for 2 hr at room temperature. Following this, the slices underwent the same treatment and imaging procedure as INS-1(832/13) cells described above. The insulin antibody was purchased from Proteintech (1:1000, 66198-1-Ig, Wuhan,

China). The specificity of the EP receptor antibodies was validated in INS-1(832/13) cells using knock-downs (*Figure 5—figure supplement 1*).

## Electrophysiology

Whole-cell potassium currents in HEK293T cells and β-cells were recorded using a Multiclamp 700B amplifier (Molecular Devices, CA, USA). The extracellular solution contained (in mM): 140 NaCl, 2.5 KCl, 10 glucose, 2.5 CaCl$_2$, 10 HEPES, and 1 MgCl$_2$, pH 7.4 adjusted with NaOH. The pipette (2–3 MΩ) solution contained (in mM): 135 K-gluconate, 10 KCl, 1 CaCl$_2$, 1 MgCl$_2$, 10 HEPES, 2 Mg-ATP, and 10 EGTA, pH 7.3 adjusted with KOH. Whole-cell potassium currents were sampled at 10 kHz and filtered at 2 kHz. For β-cell AP recordings, the extracellular solution contained (in mM): 119 NaCl, 4.7 KCl, 2 CaCl$_2$, 1.2 MgSO$_4$, 1.2 KH$_2$SO$_4$, 10 HEPES, pH 7.4 adjusted with NaOH. The pipette solution contained (in mM): 90 KCl, 50 NaCl, 1 MgCl$_2$, 10 EGTA, 10 HEPES, pH 7.3 adjusted with KOH. All of the electrophysiological recordings mentioned above were performed at room temperature.

## Isolation of primary mouse pancreatic islets

Pancreatic islets were isolated from 7-week-old wild-type and *Kcnb2*-KO male mice as previously described (*Lernmark, 1974*). Briefly, mice were euthanized, and a collagenase XI solution (0.5 mg/mL, C7657, Sigma, St. Louis, MO, USA) was injected into the pancreas via the common bile duct with approximately 3 mL per mouse. The intact pancreas was carefully dissected, and digested for 16 min at 37°C. Afterward, 20 mL of HBSS solution was added, and the mixture was filtered through a 60 µm sieve. After brief centrifugation, a density gradient centrifugation system was established using Histopaque-1077 (10771, Sigma, St. Louis, MO, USA), Histopaque-1119 (11191, Sigma, St. Louis, MO, USA), and HBSS solution for pancreatic islet separation and purification. The islets were manually picked under a stereo microscope and cultured in RPMI 1640 with 10% fetal bovine serum, 1% penicillin-streptomycin solution. The time between islet isolation and the experiment typically ranged from 24 to 48 hr.

## Intraperitoneal glucose tolerance test

Male mice were fasted overnight (16 hr) and then injected with D-glucose at a dose of 2.5 g/kg of body weight (intraperitoneal). Blood samples were collected from the tail vein before glucose injection (t = 0) and 15, 30, 60, 90, and 120 min after the glucose administration (N = 4 animals per group). A 20% glucose solution tailored to each mouse's actual body weight (following the standard of 10 µL/g) was intraperitoneally injected. Glucose levels were measured using a glucometer (Elite, Bayer, GER), following the manufacturer's recommendations. PGE2 (500 µg/kg of body weight) or saline was intraperitoneally injected 1 hr before the glucose administration. All animals were grouped based on sex, age, and genotype. Mice with the same sex, age, and genotype were randomly assigned to different control or treatment groups. The sampling of blood glucose level detection process was carried out in a double-blind fashion.

## Insulin secretion assays

For GSIS in INS-1(832/13) cells: INS-1(832/13) cells were cultured in 24-well plates, grown to approximately 90% confluency, washed with PBS, and preincubated for 1 hr in KRB solution (in mM: 120 NaCl, 4.7 KCl, 2.5 CaCl$_2$, 1.2 KH$_2$PO$_4$, 1.2 MgSO$_4$, 25 NaHCO$_3$, 10 HEPES) without glucose. The cells were incubated in the KRB solution for an additional 1 hr in the presence of either 2.8 mM (2.8 G) or 16.7 mM (16.7 G) glucose with or without PGE2, as indicated. Insulin levels in KRB solution were quantified by the Insulin ELISA Kit (90080, Crystal Chem, Chicago, IL, USA). For GSIS in islets: islets from five WT or KO male mice aged 7 weeks were used. Ten islets per mouse were preincubated in KRB buffer (supplemented with 0.025% BSA, pH 7.4) for 1 hr. Subsequently, the medium was removed, followed by sequential treatment with KRB solution (containing 16.7 mM glucose with or without PGE2) for 1 hr. Insulin in the supernatant was quantified using the same ELISA kit as above.

## Generation of *Kcnb2* knockout mice

The Kv2.2 channel coding gene, *Kcnb2* (Accession: NM_001098528.3), underwent conventional knockout using CRISPR-Cas9 gene editing. Two guide RNAs (gRNAs) were designed to target the *Kcnb2* gene. The sequences for the gRNAs are as follows: gRNA1: GAGAGTTAAGATCAACGTAG

; gRNA2: AACTCGTCCGTGGCTGCAAA. The Cas9 protein was prepared and complexed with the two synthesized gRNAs to form ribonucleoprotein complexes. Ribonucleoprotein complexes were microinjected to fertilized C57BL/6 mouse oocytes to induce double-strand breaks at the *Kcnb2* gene target sites. The microinjected embryos were cultured in vitro to the appropriate developmental stage and then transferred into the oviducts or uteruses of pseudo pregnant recipient female mice. The recipient females gave birth to $F_0$ offspring. Genotyping of $F_0$ mice: DNA was extracted from the tail tips of the $F_0$ mice. PCR was performed using two pairs of primers to amplify the regions flanking the target sites of the *Kcnb2* gene. The primer sequences are as follows: F1: TGATGTGGCGATGCCTATTCC; R1: TTCCCACAGACTAACACTTACGG; R2: TCTTCTGATGGTATCTGGCTTGG. The PCR products were purified and sequenced to confirm the knockout of the *Kcnb2* gene. Mice confirmed to have the desired *Kcnb2* gene knockout were bred to produce a stable line of *Kcnb2* conventional knockout mice for further studies. The *Kcnb2* knockout mouse was created by Cyagen Biosciences (Suzhou, China). All animals were accommodated in specific pathogen-free Fudan University facilities following a 12 hr light-dark cycle.

## Data analysis

The electrophysiological data were analyzed using Clampfit 10.7 (Molecular Devices, CA, USA). Quantitative analysis of the western blot experiments was performed with ImageJ (v1.53, NIH, USA). Data are given as the mean ± SEM. Two-tailed paired or unpaired t-test was used to compare two samples, and one-way ANOVA with Bonferroni post hoc test was employed for the comparison of multiple samples. A p-value<0.05 was considered significant. All statistical analyses were performed using GraphPad Prism (v9.4, GraphPad Software Inc, USA).

## Acknowledgements

This work was supported by the National Key Research & Development Program of China (2022YFC3602700 & 2022YFC3602702), the Science and Technology Innovation 2030 – Brain Science and Brain – Inspired Intelligence Project (2021ZD0201301), the Natural Science Foundation of Shanghai (23ZR1425900), the National Natural Science Foundation of China (31771282; 32200797).

## Additional information

### Funding

| Funder | Grant reference number | Author |
|---|---|---|
| National Key Research and Development Program of China | 2022YFC3602700 | Changlong Hu |
| Natural Science Foundation of Shanghai Municipality | 23ZR1425900 | Changlong Hu |
| National Natural Science Foundation of China | 31771282 | Changlong Hu |
| National Natural Science Foundation of China | 32200797 | Zhaoyang Li |
| National Key Research and Development Program of China | 2022YFC3602702 | Changlong Hu |
| Brain Science and Brain – Inspired Intelligence Project | 2021ZD0201301 | Changlong Hu |

The funders had no role in study design, data collection and interpretation, or the decision to submit the work for publication.

## Author contributions

Chengfang Pan, Formal analysis, Investigation, Writing – original draft, Writing – review and editing; Ying Liu, Investigation, Methodology; Liangya Wang, Wen-Yong Fan, Yunzhi Ni, Xuefeng Zhang, Di Wu, Chenyang Li, Investigation; Jin Li, Writing – review and editing; Zhaoyang Li, Conceptualization, Investigation; Rui Liu, Supervision, Writing – review and editing; Changlong Hu, Conceptualization, Supervision, Writing – original draft, Writing – review and editing

## Author ORCIDs

Jin Li ⓘ https://orcid.org/0000-0002-7957-1476
Zhaoyang Li ⓘ http://orcid.org/0000-0002-0583-1335
Changlong Hu ⓘ https://orcid.org/0000-0002-8609-8947

## Ethics

This study does not involve human participants directly. The human pancreatic samples used in this study were sourced from the Biobank of Endocrine and Metabolic Diseases, with written informed consent obtained from all patients. These samples did not include any personally identifiable information. All research procedures were approved by the Ethics Committee of Huashan Hospital, Fudan University (Approval No. 2016-395), and were conducted in accordance with the principles outlined in the Declaration of Helsinki.

All studies were carried out in strict accordance with the recommendations in the Guide for the Care and Use of Laboratory Animals of the National Institutes of Health. Animal experimental protocols were all approved by the Committee on the Ethics of Animal Experiments of Fudan University.

Reviewer #2 (Public review): https://doi.org/10.7554/eLife.97234.3.sa1
Author response https://doi.org/10.7554/eLife.97234.3.sa2

---

# Additional files

## Supplementary files

MDAR checklist

## Data availability

All data generated or analysed during this study are included in the manuscript and supporting files. The raw data used for statistical analysis are shown in the Source Data.

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
