## [Editor Report · eLife Assessment]

The study presents **valuable** findings on the molecular mechanisms of glucose-stimulated insulin secretion from pancreatic islets, focusing on the main regulatory elements of the signaling pathway in physiological conditions. While the evidence supporting the conclusions is **solid**, the study can be strengthened by the use of a beta cell line or knockout mice. The work will be of interest to cell biologists and biochemists working on diabetes.

---

## [Referee Report · Reviewer #2 (Public review)]

The authors identified new target elements for prostaglandin E2 (PGE2) through which insulin release can be regulated in pancreatic beta cells under physiological conditions. In vitro extracellular exposure to PGE2 could directly and dose-dependently inhibit the potassium channel Kv2.2. In vitro pharmacology revealed that this inhibition occurs through the EP2/4 receptors, which activate protein kinase A (PKA). By screening specific sites of the Kv2.2 channel, the target phosphorylation site (S448) for PKA regulation was found. The physiological relevance of the described signaling cascade was investigated and confirmed in vivo, using a Kv2.2 knockdown mouse model.

The strength of this manuscript is the novelty of the (EP2/4-PKA-Kv2.2 channel) molecular pathway described and the comprehensive methodological toolkit the authors have relied upon.

The introduction is detailed and contains all the information necessary to place the claims in context. Although the dataset is comprehensive and a logical lead is consistently built, there is one important point to consider: to clarify that the described signaling pathway is characteristic of normal physiological conditions and thus differs from pathological changes. It would be useful to carry out basic experiments in a diabetes model (regardless of in mouse or rat even).

Comments on revisions:

The authors addressed my comments sufficiently. I have no additional questions to clarify.

---

## [Author Response]

The following is the authors’ response to the original reviews.

**Public Reviews:**

**Reviewer #1 (Public Review):**
Summary:This study investigated the mechanism by which PGE2 inhibits the release of insulin from pancreatic beta cells in response to glucose. The researchers used a combination of cell line experiments and studies in mice with genetic ablation of the Kv2.2 channel. Their findings suggest a novel pathway where PGE2 acts through EP2/EP4 receptors to activate PKA, which directly phosphorylates a specific site (S448) on the Kv2.2 channel, inhibiting its activity and reducing GSIS.Strengths:- The study elegantly demonstrates a potential pathway connecting PGE2, EP2/EP4 receptors, PKA, and Kv2.2 channel activity, using embryonic cell line.- Additional experiments in INS1 and primary mouse beta cells with altered Kv2.2 function partially support the inhibitory role of PGE2 on GSIS through Kv2.2 inhibition.Weaknesses:- A critical limitation is the use of HEK293T cells, which are not pancreatic beta cells. Functional aspects can differ significantly between these cell types.- The study needs to address the apparent contradiction of PKA activating insulin secretion in beta cells, while also inhibiting GSIS through the proposed mechanism.- A more thorough explanation is needed for the discrepancies observed between the effects of PGE2 versus Kv2.2 knockdown/mutation on the electrical activity of beta cells and GSIS.

Thank you for your positive evaluation and constructive feedback on our study. We appreciate the concern regarding the use of HEK293T cells, which are not pancreatic beta cells and may exhibit functional differences. In response, we have repeated our key experiments using INS1 cells and primary mouse beta cells, which are more representative of the native beta cell environment. These additional experiments confirm our hypothesis and further support the role of Kv2.2 in PGE2-induced inhibition of GSIS. In beta cells, glucose-induced PKA activation is highly localized. As a result, while some PKA pathways promote insulin secretion, others may inhibit it. To directly demonstrate that PGE2-induced PKA phosphorylation of Kv2.2 is involved in the inhibitory effect on GSIS, we overexpressed the S448A mutant Kv2.2 channel in INS-1(832/13) cells. Our results show that Kv2.2-S448A channels significantly attenuate the inhibitory effect of PGE2 on GSIS, further supporting the critical role of Kv2.2 phosphorylation at S448. These data have been added to the revised Figure 7C.

**Reviewer #2 (Public Review):**
The authors identified new target elements for prostaglandin E2 (PGE2) through which insulin release can be regulated in pancreatic beta cells under physiological conditions. In vitro extracellular exposure to PGE2 could directly and dose-dependently inhibit the potassium channel Kv2.2. In vitro pharmacology revealed that this inhibition occurs through the EP2/4 receptors, which activate protein kinase A (PKA). By screening specific sites of the Kv2.2 channel, the target phosphorylation site (S448) for PKA regulation was found. The physiological relevance of the described signaling cascade was investigated and confirmed in vivo, using a Kv2.2 knockdown mouse model.The strength of this manuscript is the novelty of the (EP2/4-PKA-Kv2.2 channel) molecular pathway described and the comprehensive methodological toolkit the authors have relied upon.The introduction is detailed and contains all the information necessary to place the claims in context. Although the dataset is comprehensive and a logical lead is consistently built, there is one important point to consider: to clarify that the described signaling pathway is characteristic of normal physiological conditions and thus differs from pathological changes. It would be useful to carry out basic experiments in a diabetes model (regardless of whether this is in mice or rats).

Thank you for your positive evaluation and insightful comment. We have clarified in the Discussion section that our findings pertain specifically to physiological conditions. We acknowledge the importance of investigating the signaling pathway in a pathological context and plan to conduct experiments using a diabetes model in future studies to explore how this pathway may differ under such conditions.

**Recommendations for the authors:**

**Reviewer #1 (Recommendations For The Authors):**
(1) Figure 3A-C: PKA activation regulates different functional aspects in beta cells and HEK293T cells. It is well known that PKA activation enhances insulin secretion in beta cells, therefore the mechanisms that allow the same pathway at the same time to inhibit GSIS are not clear and should be addressed by experiments in beta cells.

Thank you for your insightful comment. Specificity and versatility in cAMP-PKA signaling are governed by the spatial localization and temporal dynamics of the signal. In beta cells, glucose-induced PKA activation is highly localized (Tengholm and Gylfe, 2017). As a result, while some PKA pathways promote insulin secretion, others may inhibit it. For example, a global increase in cAMP, such as through treatment with Db-cAMP, can simultaneously activate both stimulatory and inhibitory PKA pathways, reflecting a more integrated, complex response. In previous studies, 1 mM Db-cAMP was shown to enhance GSIS in INS-1 cells (Dezaki et al., 2011). We observed that 1 mM Db-cAMP increased GSIS, but lower concentrations (10 mM) decreased GSIS (as shown in Author response image 1). These findings suggest that not all PKA signaling events increase GSIS. To further investigate the role of PGE2-induced PKA phosphorylation of Kv2.2 in the inhibition of GSIS, we overexpressed the S448A mutant of Kv2.2 in INS-1 (832/13) cells. Our results showed that the Kv2.2-S448A mutant significantly attenuated the inhibitory effect of PGE2 on GSIS. These new data have been incorporated into the revised Figure 7C.

**Author response image 1. sa2fig1:** Effect of Db-cAMP on GSIS in INS-1 cells. Statistics for the effect of different concentrations of Db-cAMP on GSIS in INS-1(832/13) cells. One-way ANOVA with Bonferroni post hoc test. *p < 0.05; ***p < 0.001; ****p < 0.0001; n.s., not significant.

(2) Figure 3G: One would expect that the phospho-mimetic mutation, S448D, will have an opposite effect to S448A and a similar effect as PGE2 or PKA activator in Figure 3B. There is no explanation by the authors for having the same effect in S448A and S448D.

Thank you for your thoughtful comment. Indeed, the S448D mutation exhibited a similar effect to PGE2 on Kv2.2 channels, as we observed significantly smaller currents compared to wild-type Kv2.2 (Figure 3F). The S448D mutation mimics the phosphorylated state of S448, and since PGE2 regulates Kv2.2 channels by phosphorylating this residue, it has no further effect on the S448D mutant (Figure 3G). In contrast, the S448A mutation prevents phosphorylation at this site, which explains why PGE2 has no effect on the currents of S448A mutant Kv2.2 channels (Figure 3H). These results confirm that PGE2 modulates Kv2.2 channels specifically through phosphorylation of S448, as evidenced by the lack of effect on both the S448A and S448D mutants.

(3) Figure 4E: Since both PGE2 and Kv2.2 KD inhibit the activity of the channel, it doesn't definitively prove whether PGE2 acts through Kv2.2 in INS-1 cells. A complementary experiment should be done in which overactivation of Kv2.2 rescues the effect of PGE2. For example, with the S448A form of the channel.

We appreciate your comment and valuable suggestion. Knockdown of Kv2.2 abrogated the inhibitory effect of PGE2 on I_K_ currents in INS-1 cells (Figure 4E and F), which strongly indicates that PGE2 acts through Kv2.2. While we agree that the suggested complementary experiment with Kv2.2 overactivation (e.g., using the S448A mutant) could provide additional insights, we believe the current data sufficiently support our conclusion, as the knockdown of Kv2.2 eliminates the observed PGE2 effect, providing direct evidence of the channel's involvement.

(4) Figure 5C: This result requires further explanation. If PGE2 downregulates Kv2.2 activity and has an inhibitory effect on GSIS, why does Kv2.2 KD have the opposite effect?

The knockdown of Kv2.2 (Fig. 5C) reduced action potential (AP) firing rates compared to the scramble control (Fig. 5B), which is expected because Kv2.2 is critical for maintaining AP firing. When Kv2.2 is knocked down, the reduced AP firing diminishes the system’s responsiveness to further modulation by PGE2. This is because PGE2 exerts its effects primarily through Kv2.2 channels. Therefore, in the Kv2.2 knockdown condition, PGE2 does not exert an additional inhibitory effect on AP firing rates, as the channels critical for its action are already impaired.

(5) Figure 5D - The EP1-EP4 receptor antibodies should be validated at least in INS-1(832/13) cells using knockdowns.

Thank you for your suggestion. We have validated the EP1-EP4 receptor antibodies in INS-1(832/13) cells using knockdown experiments. The validation results, including confirmation of specificity and knockdown efficiency, are provided in Supplemental Figure S2.

(6) Figure 7B - These experiments don't necessarily prove that PGE2 acts directly through Kv2.2 inhibition. Using the S448A mutation in these experiments could prove this point.

Thank you for this valuable suggestion. We have now overexpressed the S448A mutant Kv2.2 channels in INS-1(832/13) cells, and the results demonstrate that Kv2.2-S448A channels significantly reduce the inhibitory effect of PGE2 on GSIS. These new data have been incorporated into the revised Figure 7C.

**Reviewer #2 (Recommendations For The Authors):**
(1) Deficiencies and inaccuracies in the description of the methods (animal numbers, name of vendors, abbreviations) and the typos in the figures (axis label) require correction.

Thank you for pointing this out. We have carefully reviewed the manuscript and the figures, making the necessary corrections to address the deficiencies in the methods section and the typos in the figure axis labels.

(2) Reducing the number of figures (Figures 7/C-E: knockout mouse line test and Figure1/HEK cell experiments could be part of supplementary) and paragraphs would make the manuscript more compact and powerful. It would also ease its reading for non-experts.

Thank you for your suggestion. We have moved Figures 7C-E to the supplementary data (Supplemental Figure S1) to streamline the main manuscript.

(3) Multiple immunostainings for EP receptors in insulinoma cells or pancreatic islets would be representative.

Due to the rabbit-derived nature of the antibodies (EP1, EP2, EP4), performing multiple immunostainings on the same samples is not feasible due to potential cross-reactivity. However, the immunohistochemistry images demonstrate that each antibody labels more than 90% of the cells, indicating that β-cell express different subtypes of EP receptors simultaneously.

(4) The antagonists chosen (AH6809, AH23848) are non-specific. Experiments should be re-run (at least some) under more stringent conditions.

Thank you for your suggestion. AH6809 and AH23848 are well-documented, widely used antagonists in the literature. To further strengthen our findings, we have included additional, widely-used antagonists: the EP2-specific antagonist TG4155 and the EP4-specific antagonist GW627368. The results obtained with these new antagonists were consistent with those observed using AH6809 and AH23848. These updated data are now included in the revised Figure 4I and 4J.

(5) It would be very helpful to indeed emphasise that this work is for physiological conditions and that it is (or is not) modified in diabetes. Maybe even irrelevant for diabetes (?). This needs to be clarified and supported by data even if one could assume the authors intend to have a follow-up entirely dedicated to pathological changes, perhaps.

Thank you for this insightful comment. We have clarified in the Discussion that our findings are specific to physiological conditions. To address this point, we have added the following statement:

‘Importantly, our findings pertain to physiological conditions. While we demonstrate the inhibitory effects of PGE2 on Kv2.2 channels in normal b-cells, the role of this pathway under diabetic conditions remains to be investigated and will be the focus of future studies.’

Dezaki K, Damdindorj B, Sone H, Dyachok O, Tengholm A, Gylfe E, Kurashina T, Yoshida M, Kakei M, Yada T (2011) Ghrelin attenuates cAMP-PKA signaling to evoke insulinostatic cascade in islet beta-cells. Diabetes 60:2315-2324.

Tengholm A, Gylfe E (2017) cAMP signalling in insulin and glucagon secretion. Diabetes Obes Metab 19 Suppl 1:42-53.